# Thymoquinone Antifungal Activity against *Candida glabrata* Oral Isolates from Patients in Intensive Care Units—An In Vitro Study

**DOI:** 10.3390/metabo13040580

**Published:** 2023-04-21

**Authors:** Noura Nouri, Shahla Roudbar Mohammadi, Justin Beardsley, Peyman Aslani, Fatemeh Ghaffarifar, Maryam Roudbary, Célia Fortuna Rodrigues

**Affiliations:** 1Department of Medical Mycology, Faculty of Medical Sciences, Tarbiat Modares University, Tehran 14115111, Iran; 2Sydney Institute for Infectious Diseases, University of Sydney, Sydney, NSW 2145, Australia; 3Westmead Hospital, NSW Health, Sydney, NSW 2145, Australia; 4Department of Parasitology and Mycology, Faculty of Medicine, Aja University of Medical Sciences, Tehran 1411718541, Iran; 5Department of Parasitology and Entomology, Faculty of Medical Sciences, Tarbiat Modares University, Tehran 14115111, Iran; 6Department of Parasitology and Mycology, School of Medicine, Iran University of Medical Sciences, Tehran 1449614535, Iran; 7TOXRUN—Toxicology Research Unit, Cooperativa de Ensino Superior Politécnico e Universitário—CESPU, 4585-116 Gandra PRD, Portugal; 8LEPABE—Laboratory for Process Engineering, Environment, Biotechnology and Energy, Faculty of Engineering, University of Porto, Rua Dr. Roberto Frias, 4200-465 Porto, Portugal; 9ALiCE—Associate Laboratory in Chemical Engineering, Faculty of Engineering, University of Porto, Rua Dr. Roberto Frias, 4200-465 Porto, Portugal

**Keywords:** thymoquinone, *Candida glabrata*, *EPA6* and *EPA7*, antifungal susceptibility, oral candidiasis

## Abstract

The number of *Candida* spp. infections and drug resistance are dramatically increasing worldwide, particularly among immunosuppressed patients, and it is urgent to find novel compounds with antifungal activity. In this work, the antifungal and antibiofilm activity of thymoquinone (TQ), a key bioactive constituent of black cumin seed *Nigella sativa* L., was evaluated against *Candida glabrata,* a WHO ‘high-priority’ pathogen. Then, its effect on the expression of *C. glabrata EPA6* and *EPA7* genes (related to biofilm adhesion and development, respectively) were analyzed. Swab samples were taken from the oral cavity of 90 hospitalized patients in ICU wards, transferred to sterile falcon tubes, and cultured on Sabouraud Dextrose Agar (SDA) and Chromagar *Candida* for presumptive identification. Next, a 21-plex PCR was carried out for the confirmation of species level. *C. glabrata* isolates underwent antifungal drug susceptibility testing against fluconazole (FLZ), itraconazole (ITZ), amphotericin B (AMB), and TQ according to the CLSI microdilution method (M27, A3/S4). Biofilm formation was measured by an MTT assay. *EPA6* and *EPA7* gene expression was assessed by real-time PCR. From the 90 swab samples, 40 isolates were identified as *C. glabrata* with the 21-plex PCR. Most isolates were resistant to FLZ (*n* = 29, 72.5%), whereas 12.5% and 5% were ITZ and AMB resistant, respectively. The minimum inhibitory concentration (MIC_50_) of TQ against *C. glabrata* was 50 µg/mL. Importantly, TQ significantly inhibited the biofilm formation of *C. glabrata* isolates, and *EPA6* gene expression was reduced significantly at MIC_50_ concentration of TQ. TQ seems to have some antifungal, antibiofilm (adhesion) effect on *C. glabrata* isolates, showing that this plant secondary metabolite is a promising agent to overcome *Candida* infections, especially oral candidiasis.

## 1. Introduction

Oropharyngeal candidiasis (OPC) is an opportunistic mucosal infection caused by *Candida* species [1], and a frequent infection among immunocompromised patients, such as those who received organ transplantation, submitted to long-term antibiotic therapy, cancer and hematological malignancy patients undergoing chemotherapy or taking immunosuppressive drugs, or HIV-positive individuals [2]. OC is characterized by the presence of creamy, white plaques on the tongue and buccal mucosa that generally leave a raw, painful, and ulcerated surface when they are scraped [3]. High-risk patients suffering from OC not only experience a lower quality of life, but may develop significant life-threatening invasive candidiasis, where the causative agents enter the bloodstream and cause candidemia [4]. 

*Candida albicans* is still the most common causative agent of OPC; but the number of non-albicans *Candida* (NAC) species (such as *Candida glabrata* (now known as *Nakaseomyces glabrata*), *Candida krusei* (*Pichia kudriavzevii*), *Candida parapsilosis*, *Candida tropicalis*, *Candida kefyr*, *Candida dubliniensis* or *Candida guilliermondii* (*Meyerozyma guilliermondii*), and *Candida orthopsilosis*) has risen worldwide in clinical settings [5,6]. Unfortunately, the majority of these species either are intrinsically less responsive to azole drugs (e.g., fluconazole, itraconazole, clotrimazole, and econazole) or acquire a high level of resistance to antifungal agents [7]. *Candida glabrata* is a commensal NAC yeast living in human mucosal surfaces (e.g., the mouth esophagus and intestine), but it can easily turn into a pathogen, especially in immunocompromised individuals, leading to a high rate of morbidity and mortality [8,9,10,11]. This species is frequently resistant to antifungal agents [12], owing a its capacity to produce biofilms (microbial complex communities), which lead to recurrent infections and higher rates of therapeutic failure [13,14,15]. It was designated as a ‘high-priority pathogen’ in the 2022 WHO fungal priority pathogen list [16]. 

The ability to adhere to epithelial cells of the oral cavity is a key initial step in the pathogenicity of an OC infection [17]. *C. glabrata* biofilms have a complex network of genes, which contribute to several pathogenesis and virulence features, such as the ability to adhere to surfaces [18]. Among the adhesion genes that contribute to pathogenesis and biofilm formation in *C. glabrata*, two of the most crucial and relevant ones are epithelial adhesion genes *EPA6* and *EPA7* [19,20,21,22,23]. The expression of *EPA6* and *EPA7* genes has also been shown to be highly induced in biofilms. *Epa6* is also known to mediate the binding of *C. glabrata* to the human extracellular matrix protein, fibronectin, which clearly indicates the importance of these genes in the biofilm maturation and, consequently, in the progress of oral *Candida* disease [24]. 

With the widespread use (and misuse) of antifungals, drug-resistant, and uncommon *Candida* species associated with oral infections (e.g., OPC) have considerably increased [25,26,27]. As conventional drugs become less effective, there is an urgent need for new approaches, such as active natural products from medicinal plants, which can act as novel and effective therapeutic strategies against pathogens [28]. 

This is the case of *Nigella sativa* L. seeds, which have been used in traditional folk medicine all over the world for over 2000 years. Indeed, *N. sativa* L. is among the top-ranked herbal medicines based on current evidence. Their seeds contain fixed and essential oils, proteins, alkaloids and saponins associated with several bioactivities. The quinine components, especially thymoquinone (TQ) (a major bioactive component of the essential oil), 2-methyl-5-isopropyl-1-benzoquinone, is responsible for the pharmacological effects of black seed, such as antihistaminic, hypoglycemic, antibacterial, antihypertensive, anti-inflammatory, and immune-enhancing effects. TQ is frequently used in the Middle East, North Africa, and South Asia. Furthermore, TQ has been reported to exhibit an interesting antifungal activity [29,30,31]. For instance, *N. sativa* L. has demonstrated a significant fungicidal effect against *Fusarium oxysporum* and has been revealed to be a natural, environmentally friendly fungi toxicant [32]. In a similar study in Iran, *N. sativa* L. TQ had strong an antifungal effect on *Trichophyton mentagrophytes*, *Microsporum canis,* and *Microsporum gypseum* (several pathogenic dermatophyte strains) [31]. Interestingly, a study indicated that at higher TQ concentrations, with a steep dose–effect relationship, the growth of a clinical isolate of *Fusarium solani* was effectively inhibited as compared with that of amphotericin B, which gave shallow dose–effect relationship [33]. However, the synergistic activity of TQ with antifungal drugs represents a promising finding. This is the example of a recent study, in which the results showed that TQ and nystatin had a synergistic effect on standard strains of *Candida* [34].

In view of the past and present literature, the current study aimed to test the antifungal and antibiofilm bioactivity of TQ, as an alternative or adjunctive therapeutic agent against 40 oral clinical isolates of *C. glabrata* from ICU hospitalized patients who underwent treatment in a clinical setting, in comparison to common antifungal drugs, fluconazole, itraconazole, and amphotericin B. This had not been assessed against clinical isolates of *C. glabrata* before. In addition, an alternative comprehensive multiplex PCR assay, 21-plex PCR, was evaluated for its identification accuracy. Moreover, to our knowledge, for the first time, an additional mechanism of action of TQ, possibly related to the downregulation of the expression of *C. glabrata EPA6* and *EPA7* genes (directly related to biofilm adhesion) was also explored.

## 2. Materials and Methods

### 2.1. Study Design

In this study, 90 hospitalized patients in intensive care unit (ICU) wards from teaching hospitals in Iran were enrolled from December 2017 to May 2018. Men and women >18 years old with clinical signs of oral candidiasis were included.

### 2.2. Thymoquinone and Antifungal Drugs

Aliquots of 400 uL/mL were prepared in dimethyl sulfoxide (DMSO, Sigma-Aldrich, USA) and kept in storage at −20 °C. Thymoquinone (Sigma-Aldrich, USA) was used at concentrations of 400–0.78 µg/mL [35], Amphotericin B (AMB) (Sigma Chemical Corporation, St. Louis, MO, USA) was used at concentrations of 0.016–16 μg/mL, Fluconazole (FLZ) (Pfizer, New York, NY, USA) was used at concentrations of 0.063–64 μg/mL, and Itraconazole (ITZ) (Santa Cruz Biotech, Dallas, TX, USA) was used at concentrations of 0.016–16 μg/mL.

### 2.3. Patients’ Characteristics

All patients were examined by specialist clinicians for signs and symptoms of OC. Patients that met the following criteria were considered to be OC positive: (a) presence of thrush (pseudomembranous) in the oral cavity, and (b) acquisition of positive yeast growth from a swab sample [36]. Oral mucositis was defined by observing erythematous and ulcerative lesions and all stages of mucositis (mild, moderate, and severe) were included [37]. This study was reviewed by ethical committee members of the Tarbiat Modares University, and ethical approval was granted with this number: IR.MODARES.REC.1400.109. Informed consent was obtained from all patients whose identity was anonymized to researchers by using a numerical code identifier (1–90). Demographic data, such as age, gender, underlying diseases, the history of antibiotic and antifungal therapy, immunosuppressing therapy, clinical symptoms of OC, and previous history of OC, were recorded in a questionnaire for OC-positive patients. Patients under 15 years old (both male and female) and pregnant women were excluded from this study.

### 2.4. Clinical Specimens Processing and Identification of Isolates

Specimens were taken from the tongue and buccal mucosal lesions of the symptomatic patients using sterile cotton swabs and immediately transferred into falcon tubes containing sterile phosphate-buffered saline (PBS 1×, 1*M*). Initially, samples were subjected to direct microscopic examination to observe the yeast structures, followed by culturing on Sabouraud dextrose agar medium (SDA, Sigma–Aldrich, Burlington, MA, USA), and then, were incubated at 35 °C for 48 h. After they grew, plates with more than 400 colonies were considered as positive culture, as previously described [38]. To obtain pure colonies for each species, they were streaked on CHROMagar (CHROMagar *Candida,* France) and incubated for 48 h at 35 °C for presumptive identification. All isolates were definitely identified by 21-plex PCR. For this purpose, total DNA of the *Candida* isolates was extracted with a Yeast DNA extraction kit (Poyagene Azema, Tehran, Iran), and 21-plex PCR was used to accurately identify the isolates according to a previous method [39]. PCR products were run on 2% agarose gel, stained with GelRed (BioTium, CA, USA), and visualized with a gel documentation device (Gel Doc XR+, BioRad, Hercules, CA, USA). Finally, *C. glabrata* isolates were chosen for further examinations, as the focus of attention in this study (Appendix A).

### 2.5. Antifungal Susceptibility Testing (AFST)

Antifungal susceptibility profile analysis of *C. grabrata* was carried out according to the Clinical and Laboratory Standards Institute (CLSI, M27-A3/S4) guideline [40,41]. Plates were incubated at 35 °C for 24 h, and visual data were recorded. *C. glabrata* (ATCC 90030), *C. krusei* (ATCC6258), and *C. parapsilosis* (ATCC22019) were used for quality control purposes. MIC_50_ for FLZ, ITZ, and TQ was defined as the minimum concentration of drugs to inhibit 50% of fungal growth, whereas for AMB, the MIC endpoint was considered to be the lowest concentration that inhibited 100% of fungal growth as compared with that of the drug-free control [42]. All tests were performed in triplicate, and proper positive and negative controls were used for tests. The growth of fungi was checked visually. MIC values of FLZ and ITZ were interpreted based on clinical breakpoints (CBP) [40].

### 2.6. Real-Time PCR Assay

To evaluate the effect of TQ on *EPA6* and *EPA7* gene expressions, RNA was extracted from *C. glabrata* isolates, after exposure to MIC_50_ (50 µg/mL) for 24 hr of TQ [42]. Since TQ at MIC_50_ could inhibit 50% of fungal growth, we used this concentration (50 µg/mL) to understand the effect of this concentration on the gene expression profiles in *C. glabrata.* For this purpose, the cell pellets of *C. glabrata* were suspended in 1 mL ice-cold RNX-plus solution using the total RNA extraction kit (Cinagen, Tehran, Iran). RNA concentration was measured using Nanodrop spectrophotometer (Nanodrop Technologies, Thermo-Fisher Scientific) and kept at −20 °C until use. Next, cDNA was synthesized from RNA using cDNA Synthesis kit (Parstous, Mashhad, Iran) according to the kit’s instructions using MultiScribeTM reverse transcriptase. Subsequently, synthesized cDNA was subjected to real-time PCR for evaluation of *EPA6* and *EPA7* gene expression of *C. glabrata* against specific primers sequences (Table 1). *ACT1* was chosen as a housekeeping gene. For this, a total of 25 uL of mixture for each reaction was prepared, including High Rox PCR Master Mix (SYBR™ Green, 2×), forward and reverse primers from each gene (10 pmol), deionized water (ddH_2_O), and cDNA as a template. PCR was carried out with following program: denaturation 95 °C for 5 min, elongation at 56 °C for 30 s, and extension at 72 °C for 1 min in 35 cycles for completing the reaction using the real-time PCR machine (Applied Biosystems Step OnePlus). The specific primers for *EPA6* and *EPA7* were designed using Allele ID primer design software (version 7.5). Results of the real-time PCR were analyzed using REST 2009 software (Ver. 2.0.13) [42].

### 2.7. Antibiofilm Activity of TQ

The effect of TQ on *C. glabrata* biofilm ability was determined by using (3-(4,5-dimethylthiazol-2-yl)-2,5-diphenyltetrazolium bromide (MTT) assay. In the first step, the *C. glabrata* biofilm was formed in flat-bottomed 96-well polystyrene cell culture plates with low evaporation lids (Becton Dickinson, Franklin Lakes, NJ, USA) as follows: *C. glabrata* cells were washed twice with PBS and suspended in RPMI-1640 cell culture medium (Sigma-Aldrich, Saint Louis, MO, USA) without sodium bicarbonate, supplemented with L-glutamine (Gibco, Grand Island, NY, USA). RPMI-1640 media were buffered with 0.165 M 3-morpholinopropane-1-sulfonic acid, MOPS (Nacalai Tesque, Kyoto, Japan), at pH 7.0.

*C. glabrata* cell suspension (100 µL) was adjusted to 1 × 10^5^ cells, which were dispensed into wells. The plate was covered, sealed with parafilm, and incubated for 24 h at 37 °C. After this time, cell suspension was aspirated and washed twice with PBS pH 7.4 to remove non-adherent cells. Residual PBS in the wells was removed, and 100 µL of TQ prepared in (DMSO, Sigma-Aldrich, Saint Louis, MO, USA) at a concentration of MIC_50_ was added to each well. The plate was incubated for 24 h at 37 °C. Then, 50 µL of MTT solution was prepared from MTT sodium salt (Sigma-Aldrich, Saint Louis, MO, USA) and added. The plate was covered and incubated in the dark for 3 h at 37 °C. Finally, 150 µL of the DMSO (dimethyl sulfoxide, Sigma-Aldrich, Saint Louis, MO, USA) was added into wells, and the optical density at 570 nm was determined using a microplate reader (Biotek, Winooski, VT, USA) [43]. Wells without TQ were considered to be negative controls.

## 3. Results

### 3.1. Clinical Data of the Patients

This study enrolled 90 hospitalized ICU patients with OPC. *C. glabrata* was isolated from 40 patients. Among them, 21 (52.5%) were female (age range 34–83 years) and 19 (47.5%) were male (age range 15–85 years). Several underlying diseases were identified, including Chronic obstructive pulmonary disease (COPD) (15%), cancer (32.5%), and pneumonia (15%) (Table 2). *C. glabrata* was the most common *Candida* species isolated from patients who stayed in the ICU for more than 7 days. Oropharyngeal cancer was the most predominant underlying disease among hospitalized patients with OC in the ICU (46.1%), followed by pneumonia and COPD. *Candida* species identified by 21-plex PCR in the patients were as follows: *C. glabrata* (44.4%), *C. albicans* (33.3%) and *C. krusei* (8.8%) and *C. tropicalis* (5.5%).

Moreover, we found an effect from the use of antibiotics (especially vancomycin), antifungal drugs, and immunosuppressive drugs (frequently prednisolone) during hospitalization in ICU and the presence of *C. glabrata* (compared to other species) in the oral cavities of patients (*p* = 0.008, 0.006, and 0.007, respectively). On the other hand, there was not a significant association between the presence of *C. glabrata* of oral cavity and the gender of patients (*p* = 0.101). We found a significant difference between the carriage of *C. glabrata* and the age of the infected patients; those older than 30 years old were more likely to be infected with *C. glabrata* than other species (*p* = 0.002).

### 3.2. Antifungal Susceptibility Testing (AFST) against TQ and Antifungal Drugs and the Effects of TQ on the Biofilm Formation of C. glabrata

AFST was performed against the 40 *C. glabrata* isolates according to the CLSI (M27 A3/S4) standard method. The results of the MIC for FLZ, ITZ, AMB, and TQ revealed that 72.5% (*n* = 29) *C. glabrata* were resistant to FLZ (MIC ≥ 64 µg/mL), and 27.5% (*n* = 11) isolates were sensitive and dose dependent (SDD) (MIC ≤ 32µg/mL). In addition, 12.5% (*n* = 5) of isolates were resistant to ITZ (MIC ≥ 2 µg/mL), 50% (*n* = 20) isolates were SDD (MIC 0.5–0.25 µg/mL), and 37.5% (*n* = 15) isolates were sensitive to ITZ (MIC ≤ 0.12 µg/mL). On the other hand, only 5% (*n* = 2) of *C. glabrata* were resistant to AMB (MIC ≥ 1 µg/mL), and 95% (*n* = 38) isolates were sensitive to AMB (MIC ≤ 1 µg/mL). The MIC_50_ of TQ was 50 µg/mL. The findings are represented in Table 3.

The inhibitory (antibiofilm) effect of TQ on *C.glabrata* biofilms was evaluated using an MTT assay. Importantly, the results indicate that the biofilm formation ability was reduced significantly (two times) in *C.glabrata* isolates exposed to MIC_50_ of TQ as compared with that of the isolates without treatment with TQ, which were used as a control group (*p* < 0.05).

### 3.3. Effects of TQ on the Expression of EPA6 and EPA7 Genes 

To assess the expression of biofilm-related genes of *C. glabrata* in presence of TQ, quantitative real-time PCR was carried out. Notably, the results showed that while the expression of *EPA7* remained comparable to the control (*p* = 0.066), 100% of the isolates treated with TQ (MIC_50_) showed a statistically significant downregulation of *EPA6* (*p* = 0.012).

## 4. Discussion

Although *C. albicans* is considered to be the major opportunistic fungal pathogen causing OPC, the number of *C. glabrata* cases are dramatically increasing along with rates of resistant [43,44,45,46,47,48,49,50,51,52,53,54,55]. Hence, great attention has been paid to alternative therapeutic agents such as natural components extracted from plants with antifungal activity against drug resistant yeast species [44].

In this work, we started by learning which *Candida* species are mostly involved in OPC in high-risk hospitalized patients in Iranian ICU wards. The results showed that *C. glabrata* was the most common *Candida* species recovered from the oral cavities of ICU patients, followed by *C. albicans, C. krusei*, and *C. tropicalis.* In agreement with our study, Deorukhkar et al. also reported a higher rate of NAC cases taken from various clinical samples, and *C. glabrata* was the major isolate from candidemia cases [56]. Similarly, Mushi et al. indicated that NAC were the predominant species isolated from the oral cavity of HIV-infected individuals [57]. According to demographic characteristic, several comorbidities were identified in the OPC patients (e.g., COPD and pneumonia) (Table 2). Moreover, that the use of antibacterial/antifungals (e.g., vancomycin) and immunosuppressive drugs during hospitalization in the ICU and the patients’ oral load of *C. glabrata* were directly related. This clearly highlights the role of these drugs as predisposing factors in OPC [56]. Although sex did not seem to matter, age was associated with the causative species.

*C. glabrata* has been ranked as a high-priority pathogen by the WHO [54], and the rest of the work was focused on this yeast and a bioactive component of the medicinal plant, TQ (as a potential drug). Based on the AFST findings, the high rate of FLZ-resistant *C. glabrata* has been a matter of global concern in clinical settings [23,25]. Our results showed that 72.5% of *C. glabrata* were resistant to FLZ, while 12.5% and only 5% isolates were resistant to ITZ. Similar results have been confirmed by other authors in Iran (although specifically, HIV patients) [45]. In another study, most people infected with *C*. *glabrata* recovered from OPC were resistant to FLZ; however, the rate of resistance to nystatin and miconazole was low [46]. Likewise, people with OPC caused by NAC who recovered from COVID-19 had a high level of resistance to FLZ, while AMB was the most effective drug against their isolates [47]. Additionally, 50% of *C. glabrata* isolates recovered from the oral cavity of patients who underwent dialysis had reduced susceptibility to FLZ [48]. 

The ability of these pathogens to form a biofilm is a prime virulence trait responsible for their multidrug resistance, which often leads to the failure of therapeutic strategies [49]. The use of bioactive molecules with proper antifungal activity introduces a promising opportunity to combat drug-resistant *Candida* spp. Therefore, we examined the inhibitory effect of TQ on resistant *C. glabrata* isolates. The MIC_50_ of TQ against *C. glabrata* was 50 µg/mL, revealing a good inhibition (low concentration of TQ) of the biofilm formation, which is in agreement with other similar studies. Al-Thobity et al. indicated that TQ incorporated into the acrylic resin denture base material dramatically decreased *C. albicans* adhesion [50]. Likewise, the nanoparticulated form of TQ showed more effective antifungal activity against both planktonic bacteria and a biofilm of *C. albicans* as compared to those of free TQ [51]. A study by Almshawit et al. showed that the fungicidal effect of TQ on different *Candida* species, particularly *C. glabrata,* in both planktonic bacteria and a biofilm. These authors indicated that this effect is probably related to the production of oxidative stress via generation of reactive oxygen species, glutathion level reduction, and the distribution of mitochondrial functions, causing cell death [52]. Similarly, a comparison between a free TQ and liposomal TQ showed a positive effect against both FLZ-susceptible or -resistant *C. albicans*. Liposomal TQ was the most effective one, and it imparted a high survival rate to mice infected with FLZ-susceptible and -resistant isolates of *C. albicans* [43].

Although several mechanisms of action of TQ in *Candida* have been described (as seen above), gene regulation disturbance has been poorly studied. Thus, we proceeded to real-time PCR analysis of two of the most important biofilm adhesion genes. The results showed that after TQ exposure (MIC 50 ug/mL), the level of *EPA6* gene expression in *C. glabrata* was statistically significantly reduced (i.e., downregulation). This means that the biofilm’s ability to adhere to a certain structure would be diminished. All taken together, these results show that TQ significantly inhibits the growth of resistant *C. glabrata* and has potential to halt the development of *C. glabrata* biofilms and, thus, the progression of the *C. glabrata* OPCs [53]. We remind the readers that these results (clinical data) are rather constricted by the power of this research, which was decreased by the limited number of *C. glabrata* isolates [48].

## 5. Conclusions

Biofilm formation is a key attribute for the progression of infections. As a result, there is a considerable amount of interest in natural anticandidal substances that are waiting to be discovered. Our findings showed that TQ has the ability to reduce *C.glabrata* CFUs, biofilm formation, and the adhesion of oral isolates, possibility being a molecule of interest for future studies to act a novel adjunctive agent to fight OC and other biofilm-involving infections. Previous work has been focused of the TQ anti-*Candida* mechanisms of action such as disintegration and disorganization with an amorphous nucleus, oxidative stress, or disturbing the cell membranes. Nonetheless, here, we also show that this natural compound can disturb gene regulation, decreasing biofilm adhesion to the structures and, therefore, their development.

Taken together, TQ is a promising therapeutic candidate to fight oral candidiasis, but further in vitro and in vivo investigations are needed to prove its clinical applications.

## Figures and Tables

**Table 1 metabolites-13-00580-t001:** List of *C. glabrata* primers used in real-time PCR.

Gene	Primer	Sequence
*ACT1*	Forward	5′-AGAGCCGTCTTCCCTTCCAT-3′
*ACT1*	Reverse	5′-TTGACCCATACCGACCATGA-3
*EPA6*	Forward	5′-AAAGCCTCAATGGTATGACAGAAGAC-3′
*EPA6*	Reverse	5′-CAGATGAATTTTGGAATGGGAAA-3′
*EPA7*	Forward	5′-GGCTGGCTTTCGTGCAATA-3′
*EPA7*	Reverse	5′-CGACGGACCCTTGTAAGATTGT-3′

**Table 2 metabolites-13-00580-t002:** Clinical features of the patients that participated in the ICU multi-center study.

Feature	Total *n* (%)	*p* Value
Age (age range 15–95 years mean ± SD 59 ± 11.4)	<30 5 (5.5%)	0.06
	>30 85 (94.4)	**0.002**
Gender (total *n*)	*n* = 40	0.101
Male	19 (47.5)	
Female	21 (52.5)	
Cancer Positive	13 (32.5)	
^1^ COPD Positive	6 (15)	
Pneumonia Positive	6 (15)	**0.021**
Previous use antifungal drugs	12 (30)	0.06
Antibacterial therapy	40 (100)	**0.008**
Immunosuppressive drugs	19 (47.5)	**0.006**
Duration of admission	3–10 days (>6 days)	**0.007**

Bold: statistically significant result (*p* < 0.05); ^1^ COPD (chronic obstructive pulmonary disease).

**Table 3 metabolites-13-00580-t003:** Antifungal susceptibility pattern of the *C. glabrata* isolates against antifungal drug.

TQ MIC Range 25–100		FLZ GM;10.11						ITZ GM;0.51						AMB GM;0.57				
MIC_50_	MIC_90_	MIC_50_	MIC_90_	MIC range	R ≥64	S	SDD ≤32	MIC_50_	MIC_90_	MIC range	R ≥2	SDD 0.5 ≤ MIC ≤ 0.25	S ≤0.125	MIC_50_	MIC_90_	MIC range	R ≥ 2	S ≤1
50	100	16	64	4–64	(*n* = 29, 72.5%)	0	(*n* = 11, 27.5%)	0.5	1	0.125–0.2	(*n* = 5, 12.5%)	(*n* = 20, 50%)	(*n* = 15, 37.5%)	0.5	1	0.25–2	(*n* = 2,−5%	(*n* = 38, −95%

R: resistant; S: sensitive; SDD: sensitive dose dependent; GM: geometric mean (µg/mL); MIC (µg/mL): Minimum Inhibitory Concentration for azoles mentioned. MIC_50_ was noted to be the lowest concentration of the drug that showed a 50% reduction of the growth of the test strain. MIC values of AMB were noted to be the lowest concentration of the drug that showed a 100% reduction of the growth of the test strain compared to that of a control strain grown without AMB.

## Data Availability

The data presented in this study are available on request from the corresponding author. The data are not publicly available due to the potential risk of patients or hospitals being de-identified.

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
