# Peer review of "Thymoquinone Antifungal Activity against Candida glabrata Oral Isolates from Patients in Intensive Care Units—An In Vitro Study"

_metabolites, 2023, doi:10.3390/metabo13040580_

Round 1

Reviewer 1 Report

 The manuscript entitled "Antibiofilm activity of thymoquinone against oral isolates of Candida glabrata - a multicenter ICU ward study" described the antifungal activity of thymoquinone (TQ) against C. glabrata taken from the oral cavity of 90 patients hospitalized in intensive care unit (ICU) wards from three (multicenter ???) hospitals.  This article can be published, but it requires some corrections:

1. The title corresponding to the manuscript content should read: "Thymoquinone activity against Candida glabrata oral isolates from patients of intensive care unit - an in vitro study".

2. In chapter 2.3. the Authors should complete the criteria for exclusion of patients from studies.

3. In chapter 2.5. the Authors should briefly describe how assessed antifungal susceptibility  profile of Candida glabrata (manually ? automated method ?). The Authors should explain why MIC50 was evaluated instead of MIC.

4. In chapter 2.7. the Authors should complete the information what volume of C. glabrata cell suspension was dispensed into plate wells, and in what was dissolved TQ which at a concentration MIC50 was added to each well.

5. The table 3 should be improved, because MIC50 is not applicable to amphotericin B, and for fluconazole the "sensitive" category is missing -  specify 0 (0%) for this category.

6. The Authors should assess the cytotoxicity of TQ in active concentration against C. glabrata.

7. In chapter 5 (Conclusion) the Authors should describe the potential practical use of the results of the research carried out.

Author Response

#Reviewer1

Dear Reviewer,

Thank you so much for taking the time to review our MS.

The authors made all changes in the MS according to your valuable comments in the MS file.

All changes are shown in highlighted in yellow.

The manuscript entitled "Antibiofilm activity of thymoquinone against oral isolates of Candida glabrata - a multicenter ICU ward study" described the antifungal activity of thymoquinone (TQ) against C. glabrata taken from the oral cavity of 90 patients hospitalized in intensive care unit (ICU) wards from three (multicenter ???) hospitals.  This article can be published, but it requires some corrections:

  1. The title corresponding to the manuscript content should read: "Thymoquinone activity against Candida glabrataoral isolates from patients of intensive care unit - an in vitrostudy".

Author reply: Thank you for your suggestion. According to your comment, we made some changes in the title as following: Thymoquinone Antifungal activity against Candida glabrata oral isolates from patients of intensive care unit - an in vitro study

  1. In chapter 2.3. the Authors should complete the criteria for exclusion of patients from studies.

Author reply: Male and female under 15 year’s old and pregnant women were exclusion criteria in this study. This was included in the manuscript.

  1. In chapter 2.5. the Authors should briefly describe how assessed antifungal susceptibility profile of Candida glabrata (manually ? automated method ?). The Authors should explain why MIC50 was evaluated instead of MIC.

Author reply: Thank you and apology for missing mention this before. According to (CLSI, M27-A3/S4) guideline, we read the growth of fungi in microdilution test visually, by checking the growth of isolates compared to controls in the wells (we added this in the 2.5 section). We considered MIC50, because according to standard method, the efficacy of antifungal drugs was assessed based on inhibition of 50% of fungal growth in presence of Azoles defined as clinical breakpoints (CBP) (33-35). This point also mentioned in the M&M

  1. In chapter 2.7. the Authors should complete the information what volume of C. glabratacell suspension was dispensed into plate wells, and in what was dissolved TQ which at a concentration MIC50 was added to each well.

Author reply: Thank you for your comment. C. glabrata cell suspension (100 µL) was adjusted to 1×105 cells dispensed into wells. In addition, TQ were prepared in dimethyl sulfoxide (DMSO, Sigma-Aldrich, USA) and used for MTT assay. These details have been mentioned in M&M section.

  1. The table 3 should be improved, because MIC50 is not applicable to amphotericin B, and for fluconazole the "sensitive" category is missing - specify 0 (0%) for this category.

Author reply: Thank you very much for this important comment. Regarding MIC of AMB, we apologize for the error. This was, indeed, a typo mistake in writing, and it is now corrected it in Table 3. These MIC values (of AMB) were read as the lowest concentration of the drug that showed a 100% reduction of growth of the tested strain, when compared with a control. Moreover, we included sensitive category for fluconazole in Table 3 as following: MIC: Minimum Inhibitory Concentration for Azoles mentioned as MIC50.

  1. The Authors should assess the cytotoxicity of TQ in active concentration against C. glabrata.

Author reply: Thank you for this valuable comment. In fact, the cytotoxicity of TQ on cell line has already been evaluated several times and the relevant data is available to support the safety of this component: PMC5640598; DOI: 10.4103/0973-1482.202886 and 10.1080/01635581.2013.878739. The reports indicate that there is interest in TQ applications as a drug, due to a minimum cytotoxic effect, which support our work.

  1. In chapter 5 (Conclusion) the Authors should describe the potential practical use of the results of the research carried out.

Author reply: Thank you very much for your interesting suggestion. We added some sentences which clarify the promising application of TQ based on our findings in highlighted in yellow in the conclusion.

Reviewer 2 Report

Nouri et al. studied the effect of thymoquinone on 40 isolated of C. glabrata.

2.4. “Clinical specimens processing and identification of isolates”

- the inclusion criteria mention patients with symptoms and positive culture. Were the cultures quantified? If yes, what was the significant threshold chosen?

- for definitive fungal identification 21-plex PCR was performed. Please specify shortly, the targets;

- why did the first and then the second multiplex PCR fail identification?

- while doing gel electrophoresis, did you use positive control and negative controls for C. glablata? Please insert a representative image for proof;

From 90 patients enrolled in the study, C. glabrata was isolated from 40 patients.

2.6. “Real-Time PCR assay” – the article wants to be focused on the effect of thymoquinone against the oral isolates of C. glabrata. Line 149 – mentions that the C. glabrata isolates were exposed to the MIC50 of TQ. For how much time? What concentrations?

Line 160 – Please check the time for the amplification protocol. Although the PCR program should have an initial denaturation (just the initial), 10 minutes for every denaturation step in each cycle is simply too much, as the denaturation duration should be kept as short as possible.

-        Were the primers taken from the literature or designed by the authors?

Results

-        3.1. “Clinical data of the patients” – limited impact due to the low number of patients (40)

-        3.3 “Effects of TQ on EPA6 and EPA7 genes” – these are the most interesting results (but how where the data statistically analyzed? It is not mentioned in the material and method section)

Discussion

-        Some of the results reappear in the section, simply rewritten (e.g. – lines 258-259). The Discussion section should focus more on why might that happen, without repeating the results, unnecessarily.

-        Lines 260-261 are repeating information from Introduction

Overall comments:

-        Although interesting, the article needs more work, mostly because of the: small sample size, lack of information in the material and method

-        Although TQ appears in the aim, in the title and it should be a focal point, its importance is not properly explained in the text

-        The novelty of the study is questionable. There are a lot of studies on TQ and C. glabrata ( just a few examples https://www-sciencedirect-com.am.e-nformation.ro/science/article/pii/S0944501316304475; https://link-springer-com.am.e-nformation.ro/article/10.1007/s10266-021-00667-4; https://journals.sagepub.com/doi/pdf/10.1177/1934578X1601100726 ; https://researchrepository.rmit.edu.au/esploro/outputs/doctoral/Studies-on-the-effect-of-thymoquinone-on-oxidative-stress-and-cell-wall-integrity-of-Candida-glabrata/9921861957701341 ), so what is the novelty of the study? It should be properly explained in the text.

Author Response

#Reviewer 2

Dear Reviewer,

Thank you so much for taking the time to review our MS.

The authors made all changes in the MS according to your valuable comments in the MS file.

All changes are shown in highlighted in yellow.

Nouri et al. studied the effect of thymoquinone on 40 isolated of C. glabrata.

2.4. “Clinical specimens processing and identification of isolates”

- the inclusion criteria mention patients with symptoms and positive culture. Were the cultures quantified? If yes, what was the significant threshold chosen?

Reply: Thank you for your comment.  The oral samples were cultured on SDA medium as described in this section. After growing, plates with more than 400 colony were considered as positive culture, as previously described.(ref 31)

- for definitive fungal identification 21-plex PCR was performed. Please specify shortly, the targets;

Reply: Thank you for your comment. 21-plex PCR can be used as a useful standalone technique in the routine laboratories of developing countries lacking specific and accurate identification tools, i.e. MALDI-TOF MS and Sanger sequencing, or expensive and time-consuming biochemical and phenotypic assays. The application of 21-plex PCR assays for the identification of common and rare yeasts can reduce turnaround times and costs if applied in developing countries. In fact, in the first multiplex assay the major pathogenic Candida species and in second assay the minor pathogenic Candida species and in third assay non-Candida pathogenic yeast identified For more details, please check this report: doi: 10.1099/jmm.0.001189.

- why did the first and then the second multiplex PCR fail identification?

Reply: Thank you for your comment. 21-plex PCR can be used as a useful standalone technique in the routine laboratories of developing countries lacking specific and accurate identification tools, i.e. MALDI-TOF MS and Sanger sequencing, or expensive and time-consuming biochemical and phenotypic assays. The application of 21-plex PCR assays for the identification of common and rare yeasts can reduce turnaround times and costs if applied in developing countries. In fact, in the first multiplex assay the major pathogenic Candida species and in second assay the minor pathogenic Candida species and in third assay non-Candida pathogenic yeast identified. In our study, since we identified all Candida species in the first reaction of PCR using specific primers, we did not had the necessity to perform other steps. For more details, please check this report: doi: 10.1099/jmm.0.001189.

- While doing gel electrophoresis, did you use positive control and negative controls for C. glablata? Please insert a representative image for proof;

Reply: Thank you for your comment. We used positive and negative control in every reaction of PCR. We used C. glabrata (ATCC2001) as positive control and the tube with no DNA was considered as negative control. As asked, we provided an image of the PCR in the manuscript.

From 90 patients enrolled in the study, C. glabrata was isolated from 40 patients.

2.6. “Real-Time PCR assay” – the article wants to be focused on the effect of thymoquinone against the oral isolates of C. glabrata. Line 149 – mentions that the C. glabrata isolates were exposed to the MIC50 of TQ. For how much time? What concentrations?

Author reply: Thank you. C. glabrata isolates were exposed to MIC50 of TQ (50µg/mL) for 24 hr. These details were added in this section.

Line 160 – Please check the time for the amplification protocol. Although the PCR program should have an initial denaturation (just the initial), 10 minutes for every denaturation step in each cycle is simply too much, as the denaturation duration should be kept as short as possible.

Author reply: Thank you so much for noticing this. In fact, it was a typo error. The correct time is, as the reviewer well noticed, 5 min. We edited the text.

-        Were the primers taken from the literature or designed by the authors?

Author reply: The specific primers for EPA6 and EPA7 were designed using Allele ID primer design software (version 7.5) and we ordered the primer sequences for synthesis. This data is added in this section.

Results

-        3.1. “Clinical data of the patients” – limited impact due to the low number of patients (40)

Author reply: Thank you. We clarified this point in the discussion as: We remind that these results (clinical data) are rather constricted by the power of this research, which was decreased by the limited number of C. glabrata patients (40).

-        3.3 “Effects of TQ on EPA6 and EPA7 genes” – these are the most interesting results (but how where the data statistically analyzed? It is not mentioned in the material and method section)

Author reply: Thank you for this reminder. Actually the findings of Real-time PCR were analyzed br REST 2009 software (Ver. 2.0.13), as a standard tool for analysis of gene expression. We included it into the end of this section.

Discussion

-        Some of the results reappear in the section, simply rewritten (e.g. – lines 258-259). The Discussion section should focus more on why might that happen, without repeating the results, unnecessarily.

-        Lines 260-261 are repeating information from Introduction

 Author reply: Thank you for your valuable comment. We modified discussion by removing some repeated information which already we mentioned in introduction and results to make it more informative and well classified.

Overall comments:

-        Although interesting, the article needs more work, mostly because of the: small sample size, lack of information in the material and method

Author reply: Thank you for this comment. The limited sample size has now been addressed in the discussion as a disadvantage which needs more investigation in the future studies. The lack of information in M&M was also covered in the new revision of the MS. The main findings of our work are related to a different mechanism of action of TQ in Candida biofilms and its potential use in oral candidiasis, as noted in the MS: “…TQ significantly inhibited biofilm formation of C. glabrata isolates and the EPA6 gene expression was reduced significantly at MIC50 concentration of TQ. TQ seems to have some antifungal, antibiofilm (adhesion) effect on C. glabrata isolates, showing that this plant secondary metabolite is a promising agent to overcome Candida infections, especially oral candidiasis.” We hope to have clarified and addressed the major concerns of the reviewer.

-        Although TQ appears in the aim, in the title and it should be a focal point, its importance is not properly explained in the text

-        The novelty of the study is questionable. There are a lot of studies on TQ and C. glabrata ( just a few examples https://www-sciencedirect-com.am.e-nformation.ro/science/article/pii/S0944501316304475; https://link-springer-com.am.e-nformation.ro/article/10.1007/s10266-021-00667-4; https://journals.sagepub.com/doi/pdf/10.1177/1934578X1601100726 ; https://researchrepository.rmit.edu.au/esploro/outputs/doctoral/Studies-on-the-effect-of-thymoquinone-on-oxidative-stress-and-cell-wall-integrity-of-Candida-glabrata/9921861957701341 ), so what is the novelty of the study? It should be properly explained in the text.

Author reply: Thank you for this comment. It is important to note that, most of these studies only use reference strains or an even lower number of strains. In our work, we use samples from 90 patients, and 40 C. glabrata. Furthermore, although the literature confirms the antifungal effect of TQ, mainly on reference strains or few number of isolates, here we particularly focus on C. glabrata, a (novel) WHO priority fungal infection list in 2022(ref 14).

In addition, we present, for the first time (to the best of our knowledge) the effect of TQ on EPA6 and EPA7, as underlying genes related to adhesion of C. glabrata to the host tissue, for biofilm formation and, thus, drug resistance.

Reviewer 3 Report

This manuscript by Nouri et al., Manuscript ID; metabolites-2242256, entitled “Antibiofilm activity of thymoquinone against oral isolates of Candida glabrata – a multicenter ICU ward study” describes the antibiofilm activity of thymoquinone against oral isolates of Candida glabrata. The addressed subject is interesting and within the journal's scope, but this manuscript contains some major points.

Herein are some comments to improve the manuscript:

Manuscript Title

·       From this title, it can be concluded that a clinical trial study (experimental study on humans) has been done on the patients, but this is not the case. It is recommended to delete “a multicenter ICU ward study” in the manuscript title.

Abstract

·       Line 26: Please correct “sp.” to “spp.” or change “sp.” to “species”.

·       Line 33: Please correct “SDA” to “Sabouraud Dextrose Agar”.

·       Line 34: Please correct “Candida glabrata” to “C. glabrata.

·       Line 36: Please correct “micro dilution” to “microdilution”.

·       Line 40: Please correct “MIC50” to “MIC50.

·       Line 41: Please correct “MIC50” to “MIC50.

Keywords:

·       Line 45: Please change “EPA6 and EPA7 gene expression” to “EPA6” and “EPA7”.

·       Line 45: Please add this keyword to this manuscript “antifungal susceptibility”

1. Introduction

·       Please delete this figure “Figure 1” from the manuscript. There is no need for this figure.

2. Experimental Design

2.1. Study design

·     Line 92: Please add the name of “three teaching hospitals” to the manuscript.

2.2. Thymoquinone and antifungal drugs

·       Line 96: Please delete “All compounds were purchased” from the manuscript.

·       Line 98: Based on what protocols these concentrations (400-0.78 µg/mL) were selected for thymoquinone. Please clarify it.

2.3. Patients’ characteristics

·       Line 111: Please correct “IR.REC.1400.109” to “IR.MODARES.REC.1400.109”.

·       Line 111: Please delete “(TMU)” in the manuscript.

·       Line 114: It seems that reference 29 not need for this sentence. Please delete it.

2.4 Clinical specimens processing and identification of isolates

·       Line 129: You know in 21-plex PCR, the first PCR step identifies the most prevalent Candida species [C. albicans, C. dubliniensis, C. parapsilosis, C. krusei, C. glabrata, C. tropicalis, and C. auris]. In step 1, C. glabrata was identified, Therefore There is no need to do steps 2 and 3. Please clarify it.

2.5. Antifungal susceptibility testing (AFST)

·       Line 138: Please add a relevant reference for CLSI, M27-A3/S4) guideline.

·       Line 139: You mentioned that “Plates were incubated at 35° C for 24-48 h”, in CLSI M27-A3, you must read after 24h. Please clarify it.

·       Line 139: CLSI M27-A3 recommended to used C. krusei ATCC6258 and C. parapsilosis ATCC22019 for quality control, but you used just used C. glabrata (ATCC 90030). Please explain this.

·       Line 140: It seems that you have misunderstood the MIC and MIC50.

o   We are actually talking about three different things, the minimal inhibitory concentration (MIC) itself, the MIC50, and MIC90.

1- MIC is the Minimal Concentration of an antifungal necessary to inhibit the growth of a target microorganism.

2- MIC50:The MIC50, for example, When you do antifungal susceptibility for 100 isolates, gives you the MIC, which inhibits 50% of your isolates

3- MIC90 The MIC90 gives you the MIC which inhibits 90% of the isolates of the species, tested.

Please correct this.

·       Line 141: The MIC endpoint for thymoquinone was determined based on what protocol or 50 % of inhibition fungal growth to growth control. Please clarify it in the manuscript. (100% inhibition is usually used when a drug’s mechanism is unknown.). furthermore, Almshawit et al. showed that the fungicidal effect of TQ on different Candida species, you know for fungicidal antifungal MIC was measured at 100 % of inhibition fungal growth to growth control for example amphotericin B and nystatin. Please clarify it.

2.6. Real-time PCR assay

·       Line 151: Please add a reference for this protocol.

·       You must use sub-MIC value for evaluation of gene expressions. But you used “MIC50”. Please clarify it.

·       Line 145: Is “MIC50” means MIC which inhibits 50% of the isolates? Please clarify.

·       Line 154: Please add a reference for this protocol.

·       Line 159: Please correct “ddH2O” to “ddH2O”.

·       Line 160: Please mention what type of Real-time PCR was done (SYBR™ Green or TaqMan,...).

·       Line 160: Please mention that, please add a reference for this Real-time PCR Cycle.

·        

3.2. Antifungal susceptibility testing (AFST) against TQ and antifungal drugs and the effects of TQ on the biofilm formation of C. glabrata

·       Line 221: Please correct “MIC50” to “MIC50.

·       Line 232: Please correct “C.glabrata” to “C. glabrata.

·       Line 224: Please correct “MIC50” to “MIC50.

3.3. Effects of TQ on the expression of EPA6 and EPA7 genes

·       Line 237: Please correct “C.glabrata” to “C. glabrata.

·       Line 240: Please correct “MIC50” to “MIC50.

·       Please add a graph for this analysis.

4. Discussion

·       Line 237: Please correct “C.glabrata” to “C. glabrata.

·       Line 278: Please correct “C.glabrata” to “C. glabrata.

·       Line 283: Please correct “fungicial” to “fungicidal”.

·       Line 283: Please correct “Almshawat” to “Almshawit”.

·       Line 284: Please correct “C.glabrata” to “C. glabrata.

·       Line 296: Please correct “C.glabrata” to “C. glabrata.

5. Conclusion

·       Line 302: Please correct “C.glabrata” to “C. glabrata.

Funding:

·       Line 315: Please correct “IR.REC.1400.109” to “IR.MODARES.REC.1400.109”.

Informed Consent Statement:

·       Line 317: Please correct “IR.REC.1400.109” to “IR.MODARES.REC.1400.109”.

References

·       Please correct “References 24”.

·       Please correct “References 31”.

Table 2.

·       Please add “C. glabrata” in the table legend.

·       Based on what criteria divide age into two groups. Please explain.

·       It is recommended to add age range and age mean±SD to the table.

·       Please add the full-length word “COPD” under the table.

·       Please add other risk factors of C. glabrata Oropharyngeal candidiasis in this study.

·       Please delete the negative group in the table.

Table 3.

·       Please add MIC range, Geometric mean MIC, MIC50, and MIC90 to table3.

Author Response

#Reviewer 3

Dear Reviewer,

Thank you so much for taking the time to review our MS.

The authors made all changes in the MS according to your valuable comments in the MS file.

All changes are shown in highlighted in yellow.

This manuscript by Nouri et al., Manuscript ID; metabolites-2242256, entitled “Antibiofilm activity of thymoquinone against oral isolates of Candida glabrata – a multicenter ICU ward study” describes the antibiofilm activity of thymoquinone against oral isolates of Candida glabrata. The addressed subject is interesting and within the journal's scope, but this manuscript contains some major points.

Herein are some comments to improve the manuscript:

Manuscript Title

  • From this title, it can be concluded that a clinical trial study (experimental study on humans) has been done on the patients, but this is not the case. It is recommended to delete “a multicenter ICU ward study” in the manuscript title.

Author reply: Thank you for your recommendation. The authors deleted multicenter ICU ward from the title, and changed to: Thymoquinone Antifungal activity against Candida glabrata oral isolates from patients of intensive care unit - an in vitro study.

 Line 26: Please correct “sp.” to “spp.” or change “sp.” to “species”.

Reply: It has been edited to spp.

  • Line 33: Please correct “SDA” to “Sabouraud Dextrose Agar”.

Reply: It has been edited to Sabouraud Dextrose Agar.

 Line 34: Please correct “Candida glabrata” to “C. glabrata.

Reply: It has been edited to C. glabrata.

. Line 36: Please correct “micro dilution” to “microdilution”.

Reply: It has been edited to microdilution.

 Line 40: Please correct “MIC50” to “MIC50.

Reply: It has been edited to MIC50.

 Line 41: Please correct “MIC50” to “MIC50.

Reply: It has been edited to MIC50.

Keywords:

  • Line 45: Please change “EPA6 and EPA7 gene expression” to “EPA6” and “EPA7”.
  • Line 45: Please add this keyword to this manuscript “antifungal susceptibility”

Author reply: We added antifungal susceptibilities in key word list.

  1. Introduction
  • Please delete this figure “Figure 1” from the manuscript. There is no need for this figure.

Author reply: Figure deleted.

  1. Experimental Design

2.1. Study design

 Line 92: Please add the name of “three teaching hospitals” to the manuscript.

Author reply: Thank you for this comment. We edited this sentence as following based on your comment to delete multi centers in title. In this study, 90 hospitalized patients in intensive care unit (ICU) wards from teaching hospitals in Iran.

2.2. Thymoquinone and antifungal drugs

 Line 96: Please delete “All compounds were purchased” from the manuscript.

Author reply: This sentence has been deleted.

 Line 98: Based on what protocols these concentrations (400-0.78 µg/mL) were selected for

thymoquinone. Please clarify it.

Author reply: Thank you for this valuable comment. We used this range of concentration (400-0.78 µg/mL) based on a previous report (reference 28) (also added to M&M).

2.3. Patients’ characteristics

  • Line 111: Please correct “IR.REC.1400.109” to “IR.MODARES.REC.1400.109”.

Author reply: Thank you. It has been corrected.

  • Line 111: Pleasedelete “(TMU)” in the manuscript.

Author reply: Thank you for your comment. It has been deleted in the text.

Line 114: It seems that reference 29 not need for this sentence. Please delete it.

Author reply: Thank you. This reference was deleted.

2.4 Clinical specimens processing and identification of isolates

  • Line 129: You know in 21-plex PCR, the first PCR step identifies the prevalent Candidaspecies [C. albicansC. dubliniensis, C. parapsilosisC. kruseiC. glabrata, C. tropicalis, and C. auris]. In step 1, C. glabrata was identified, Therefore There is no need to do steps 2 and 3. Please clarify it.

Author reply: Thank you for your comment. We deleted other steps of PCR in methods as all isolates identified in step 1 of PCR.                   

2.5. Antifungal susceptibility testing (AFST)

  • Line 138:Please add a relevant reference for CLSI, M27-A3/S4) guideline.

Author reply: We added reference 33 and 34 for CLSI, (M27-A3/S4) guideline in the AFST section which is highlighted.

  • Line 139:You mentioned that “Plates were incubated at 35° C for 24-48 h”, in CLSI M27-A3, you must read after 24h. Please clarify it.

Author reply: Thank you for this comment. We edited this part in M&M.

  • Line 139:CLSI M27-A3 recommended to used C. krusei ATCC6258 and C. parapsilosis ATCC22019 for quality control, but you used just used C. glabrata (ATCC 90030).

 Please explain this.

Author reply: Thank you for this comment. The authors completely agree with the reviewer and apologize for an unintended mistake, while mentioned the quality controls. The authors edited it as following in the text: C. glabrata (ATCC 90030), C. krusei (ATCC6258) and C. parapsilosis (ATCC22019) were used for quality control purposes.

  • Line 140: It seems that you have misunderstood the MIC and MIC50. We are actually talking about three different things, the minimal inhibitory concentration (MIC) itself, the MIC50, and MIC90.

1- MIC is the Minimal Concentration of an antifungal necessary to inhibit the growth of a target microorganism.

2- MIC50:The MIC50, for example, When you do antifungal susceptibility for 100 isolates, gives you the MIC, which inhibits 50% of your isolates

3- MIC90 The MIC90 gives you the MIC which inhibits 90% of the isolates of the species, tested.

Please correct this.

Author: Thank you for your explanation. Indeed, we admit that this was not clear in the manuscript. In fact, we used MIC50 for TQ and other antifungal drugs in this study, which were noted to be the lowest concentration of the drug that showed a 50% reduction of growth of the test strain. We also clarify this description in the both caption of figure 3 and 2.5 section (highlighted) (reference added doi: 10.1080/20002297.2019.160106 ).

 Line 141: The MIC endpoint for thymoquinone was determined based on what protocol or 50 % of inhibition fungal growth to growth control. Please clarify it in the manuscript. (100% inhibition is usually used when a drug’s mechanism is unknown.). furthermore, Almshawit et al. showed that the fungicidal effect of TQ on different Candida species, you know for fungicidal antifungal MIC was measured at 100 % of inhibition fungal growth to growth control for example amphotericin B and nystatin. Please clarify it.

Author reply: As we evaluated MIC50 of common antifungal drugs in this study (FLZ, ITZ (which were noted to be the lowest concentration of the drug that showed a 50% reduction of growth of the test strain however, for AMB we considered 100% inhibition of fungi growth ), in order to have a direct comparison with these drugs, we examined the antifungal activity of TQ in MIC50 (in comparison a positive control. This procedure were based on other related studied in natural components (please check references 28 and 35).

However, in the Almshawit et al study , the authors evaluated fungicide activity of TQ whereas we examined MIC of TQ not fungicide effect that inhibited 100 % of fungal growth.

2.6. Real-time PCR assay

  • Line 151: Please add a reference for this protocol.

Author reply: The reference has been added in the text.

  • You must use sub-MIC value for evaluation of gene expressions. But you used “MIC50”.

Author reply: Thank you for your comment. Since TQ at MIC50 could inhibit 50% of fungal growth, we used this concentration (50 µg/mL) to understand the direct effect of this concentration on the gene expression profiles in C.glabrata. We added this clarification in the Results section.

  • Line 145: Is “MIC50” means MIC which inhibits 50% of the isolates? Please clarify.

Author reply: Thank you. Yes, correct, it means that, at this concentration, 50% of fungal growth was inhibited, when compared to a positive control.

  • Line 154: Please add a reference for this protocol.

Author reply: The correct reference (number 34) has been added in the text.

  • Line 159: Please correct “ddH2O” to “ddH2O”.

Author reply: Thank you. We corrected it to ddH2O.

  • Line 160: Please mention what type of Real-time PCRwas done (SYBR™ Green or

 TaqMan,...).

Author reply: Thank you for your comment. It was SYBR™ Green. We added this detail in the manuscript.

  • Line 160: Please mention that, please add a reference for this Real-time PCRCycle.

Author reply: The reference has been added (number 34) in the text.

3.2. Antifungal susceptibility testing (AFST) against TQ and antifungal drugs and the effects of TQ on the biofilm formation of C. glabrata

  • Line 221: Please correct “MIC50” to “MIC50.

Author reply: Thank you. We corrected it to MIC50.

  • Line 232: Please correct “C.glabrata” to “C. glabrata.

Author reply: Thank you. We corrected it to C. glabrata.

  • Line 224: Please correct “MIC50” to “MIC50.

Author reply: Thank you. We corrected it to MIC50.

3.3. Effects of TQ on the expression of EPA6 and EPA7 genes

  • Line 237: Please correct “C.glabrata” to “C. glabrata.

Author reply: It has been corrected in the text.

  • Line 240: Please correct “MIC50” to “MIC50.

Author reply: It has been corrected in the text.

  • Please add a graph for this analysis.

Author reply: Thank you for your comment. Unfortunately, real–time graph is just available while the test is running, and we don’t usually save them, just the raw data, to the analysis (presented in our manuscript).

  1. Discussion
  • Line 237: Please correct “C.glabrata” to “C. glabrata.

Author reply: It has been corrected in the text.

  • Line 278: Please correct “C.glabrata” to “C. glabrata.

Author reply: It has been corrected in the text.

  • Line 283: Please correct “fungicial” to “fungicidal”.

  • Author reply: It has been corrected in the text.

       Line 283: Please correct “Almshawat” to “Almshawit”.

Author reply: It has been corrected in the text.

  • Line 284: Please correct “C.glabrata” to “C. glabrata.

Author reply: It has been corrected in the text.

  • Line 296: Please correct “C.glabrata” to “C. glabrata.

Author reply: It has been corrected in the text.

  1. Conclusion
  • Line 302: Please correct “C.glabrata” to “C. glabrata.

Author reply: We edited it to C. glabrata.

Funding:

  • Line 315: Please correct “IR.REC.1400.109” to “IR.MODARES.REC.1400.109”.

Informed Consent Statement:

  • Line 317: Please correct “IR.REC.1400.109” to “IR.MODARES.REC.1400.109”.

Author reply: it has been edited to IR.MODARES.REC.1400.109 in both cases.

References

      Please correct “References 24”.

       Please correct “References 31”.

Author reply: Thank you very much. All the references have been rearranged in the text and reference list according the reviewer’s comments.

Table 2.

  • Please add “C. glabrata” in the table legend.

Author reply: Thank you for this comment. This Table represent the clinical data of all patients were recruited in this study and three different Candida species were isolated from patients not just C.glabrata that these findings mentioned in the result.

  • Based on what criteria divide age into two groups. Please explain.

Author reply: Thank you for this comment. Actually, as the patients included in this study were a wide range of age, and according to an expert statistician, to do a meaningful analysis (statistical correlation) between clinical data and microbiological finding, the age was divided into less and more than 30 years old.

  • It is recommended to add age range and age mean±SD to the table.

Author reply:

Reply: done

  • Please add the full-length word “COPD” under the table.

Author reply: Thank you. We added the full length of COPD as Chronic obstructive pulmonary disease in the figure legend.

  • Please add other risk factors of C. glabrataOropharyngeal candidiasis in this study.

Author reply: Thank you for your comment. In fact, we included all clinical data like predisposing factors, that were available and reported by the medical staff of these patients.

  • Please delete the negative group in the table.

Author reply: We deleted the negative group in the table 2.

Table 3.

    Please add MIC range, Geometric mean MIC, MIC50, and MIC90 to table3.

Reply: done

Round 2

Reviewer 1 Report

Dear Authors,

thank you for taking into account my comments and making appropriate changes to the text of the manuscript , which I now recommend for printing in "Metabolites".

Reviewer 2 Report

The responses were on point and the manuscript had improved.